# Psychosocial Aspects of Living Long Term with Advanced Cancer and Ongoing Systemic Treatment: A Scoping Review

**DOI:** 10.3390/cancers14163889

**Published:** 2022-08-11

**Authors:** Evie E. M. Kolsteren, Esther Deuning-Smit, Alanna K. Chu, Yvonne C. W. van der Hoeven, Judith B. Prins, Winette T. A. van der Graaf, Carla M. L. van Herpen, Inge M. van Oort, Sophie Lebel, Belinda Thewes, Linda Kwakkenbos, José A. E. Custers

**Affiliations:** 1Radboud University Medical Center, Radboud Institute for Health Sciences, Department of Medical Psychology, 6525 Nijmegen, The Netherlands; 2School of Psychology, University of Ottawa, Ottawa, ON K1N 6N5, Canada; 3Department of Medical Oncology, Netherlands Cancer Institute, 1066 Amsterdam, The Netherlands; 4Department of Medical Oncology, Erasmus MC Cancer Institute, Erasmus Medical Center, 3015 Rotterdam, The Netherlands; 5Radboud University Medical Center, Radboud Institute for Health Sciences, Department of Medical Oncology, 6525 Nijmegen, The Netherlands; 6Radboud University Medical Center, Radboud Institute for Health Sciences, Department of Urology, 6525 Nijmegen, The Netherlands; 7School of Psychology, Sydney University, Camperdown 2050, Australia; 8Clinical Psychology, Radboud University, 6525 Nijmegen, The Netherlands; 9Radboud University Medical Center, Radboud Institute for Health Sciences, IQ Healthcare, 6525 Nijmegen, The Netherlands; 10Radboud University Medical Center, Radboudumc Center for Mindfulness, Department of Psychiatry, 6525 Nijmegen, The Netherlands

**Keywords:** psychosocial oncology, advanced cancer, systemic treatment, uncertainty, fear, hope, loss, depression, social life, quality of life, scoping review

## Abstract

**Simple Summary:**

An emerging group of advanced cancer patients are living long term on systemic treatment. However, studies examining the psychosocial impact of this prolonged cancer treatment trajectory are scarce. This scoping review summarizes findings on these psychosocial issues, as well as the terminology used to refer to these patients. Prominent psychosocial outcomes included uncertainty, anxiety, and fear of disease progression or death, hope, loss and worries about loved ones and changes in social life. These themes were not extensively investigated in research using validated psychological questionnaires. More quantitative research in this area should be conducted to further understand these psychological constructs. A large variety of terms used to refer to the patient group was observed, which calls for a uniform definition to better address this specific patient group in research and in practice. By identifying key themes and gaps in the literature, directions for future research and clinical practice can be provided.

**Abstract:**

(1) Background: Studies examining the psychosocial impact of living long term on systemic treatment in advanced cancer patients are scarce. This scoping review aimed to answer the research question “What has been reported about psychosocial factors among patients living with advanced cancer receiving life-long systemic treatment?”, by synthesizing psychosocial data, and evaluating the terminology used to address these patients; (2) Methods: This scoping review was conducted following the five stages of the framework of Arksey and O’Malley (2005); (3) Results: 141 articles published between 2000 and 2021 (69% after 2015) were included. A large variety of terms referring to the patient group was observed. Synthesizing qualitative studies identified ongoing uncertainty, anxiety and fear of disease progression or death, hope in treatment results and new treatment options, loss in several aspects of life, and worries about the impact of disease on loved ones and changes in social life to be prominent psychosocial themes. Of 82 quantitative studies included in the review, 76% examined quality of life, 46% fear of disease progression or death, 26% distress or depression, and 4% hope, while few studies reported on adaptation or cognitive aspects. No quantitative studies focused on uncertainty, loss, or social impact; (4) Conclusion and clinical implications: Prominent psychosocial themes reported in qualitative studies were not included in quantitative research using specific validated questionnaires. More robust studies using quantitative research designs should be conducted to further understand these psychological constructs. Furthermore, the diversity of terminology found in the literature calls for a uniform definition to better address this specific patient group in research and in practice.

## 1. Introduction

Since the beginning of the 2000s, the approval of new systemic treatments based on increased progression-free and overall survival rates for patients with several types of advanced cancer has led to new standards of care. As a result of these developments, there is an emerging group of patients living longer with advanced cancer while being on continuous targeted treatment or on immunotherapy.

There is extensive literature reporting the psychosocial consequences of cancer in curative and end-of-life settings [1,2,3,4]. However, studies examining the psychosocial outcomes of living long term with advanced cancer in patients receiving ongoing systemic treatment and/or follow-up are scarce [5,6,7]. As the impact on patients and their loved ones’ well-being and quality of life is expected to be considerable [8], it is crucial to examine these psychosocial outcomes to be able to support patients in their coping. Possible concerns include living long term with uncertainty about prognosis and treatment, imminent medication resistance, disease progression, and the possibility of an early death [9]. Several previous studies have observed increased anxiety around scans, referred to as ‘scanxiety’ [10], and fear of recurrence or progression and how patients dealt with this [11]. Emotional and social functioning, increased distress, and feelings of uncertainty have also been reported previously [9,11]. Patients living with metastatic GIST in long-term (partial) remission experiencing treatment side effects have been reported to have a limited daily quality of life [12]. On the other hand, feelings of hope have been reported, varying from hope for cure to hope for tumor stabilization or maintenance of quality of life [13,14]. The impact of experiencing longer survival while receiving systemic treatment is believed to be prominent for an emerging group of advanced cancer patients, but literature is scarce and focused on specific diagnoses, treatments, and outcomes. There is an urgent need to identify the most prominent themes and outcomes for this group of cancer patients in order to guide further research and to develop supportive interventions for patients based on theoretical models.

A complicating factor in addressing this patient group is the lack of a uniform term or definition. The stages in which cancer patients live are commonly categorized as curative, palliative, or terminal. While the intent of curative treatment is to make a patient cancer free, palliative treatment means that a cure is no longer possible, or at least very unlikely, and treatment is focused on prolonging life while maintaining quality of life as well as possible. The new group of patients who are living longer with advanced cancer with a poor and uncertain prognosis and who respond to one or more systemic treatment lines over longer periods of time does not fit this common classification between curative and palliative patients, and complicates the examination of this specific group because of this lack of appropriate terminology [8].

This scoping review aimed to answer the following research question: “What has been reported about the psychosocial aspects of living with advanced cancer receiving systemic treatment?” by giving a broad overview. Specific goals were to:Synthesize and categorize psychosocial outcomes that have been published in quantitative and qualitative papers;Summarize and evaluate the descriptions, definitions and terminology used in the literature to define patients with advanced cancer while receiving systemic treatment; andDetect knowledge gaps regarding psychosocial outcomes in advanced cancer patients.

## 2. Materials and Methods

This scoping review was conducted following the five stages of the framework of Arksey and O’Malley (2005) [15], incorporating the alterations of Levac et al. (2010) [16] and The Joanna Briggs Institute (2015), including: (1) developing the research question; (2) identifying eligible studies; (3) selecting studies; (4) charting the data; and (5) collating, summarizing and reporting the results. The reporting of this review followed the Preferred Reporting Items for Systematic reviews and Meta-Analyses extension for Scoping Reviews (PRISMA-ScR) reporting guidelines [17,18]. In this section, we provide a brief summary of these steps as an extensive description of the protocol has been previously published [19].

The following research question was formulated: “What has been reported about the psychosocial aspects in patients with advanced cancer receiving life-long systemic treatment?”. A comprehensive search strategy was developed (see Appendix A) and peer reviewed using the Peer Review of Electronic Search Strategy (PRESS) standard to identify empirical quantitative and qualitative studies, and non-empirical articles [20]. Studies involving adults currently diagnosed with advanced cancer and receiving lifelong or repetitive systemic treatment (i.e., chemotherapy, hormone therapy, immunotherapy and/or targeted therapy) and reporting at least one psychosocial outcome as a primary outcome were included. The search was conducted in six literature databases (MEDLINE, Embase, CINAHL, PsycINFO, Web of Science and the Cochrane Database of Systematic Review), with search limits set to English language articles, published after 2000, since the emergence of new systemic treatment options for advanced cancer, through to February 2021. First, two investigators independently reviewed the titles and abstracts of all articles identified through the search strategy. If one or both investigators deemed an article potentially eligible based on the inclusion criteria, then two investigators completed a full-text review to assess for eligibility, independently. Disagreements after full-text review were resolved by consensus, with a third investigator consulted when necessary, leading to a final set of articles for data extraction.

Two data-extraction forms were initially pilot tested, one for quantitative and mixed-methods studies, and one for qualitative studies, in a subset of 20 studies in total. Discrepancies were discussed and minor adaptations to the forms were made with input from the team. Further data extraction was independently performed by one researcher and verified by a second researcher. Quality assessment was conducted for each included study according to the ‘Standard Quality Assessment Criteria for Evaluating Primary Research Papers from a Variety of Fields’ [21] and verified by a second reviewer. All data from the included studies were charted, regardless of the study’s quality score. For papers with a study design other than quantitative, qualitative or mixed methods (e.g., commentary, review), no data were extracted and no quality assessment was conducted. However, these studies were consulted for the terminology of advanced cancer and overall scope of the literature.

Qualitative data derived from studies with a qualitative or mixed-methods design were analyzed using a thematic synthesis approach. The results sections of these studies were coded line by line in ATLAS.ti (Scientific Software Development GmbH; version 9). The first five studies were coded individually by two researchers, leading to the development of a coding tree. Discrepancies were discussed until consensus was reached. The remaining studies were coded by one researcher and reviewed by a second researcher. Codes were grouped into sub themes, which were summarized into overall themes in an iterative process in consultation with team members.

## 3. Results

### 3.1. Publication Characteristics

The database searches yielded 16,015 papers. After removing duplicates, 11,678 unique papers were screened for title and abstract, leading to the exclusion of 11,193 papers. The full texts of the remaining 485 papers were reviewed, resulting in 128 eligible articles. Thirteen additional eligible articles were identified by hand search of reference lists in eligible articles, resulting in a total of 141 publications that met the inclusion criteria and were included in the scoping review (Appendix A, Appendix A).

The 141 included papers consisted of 82 (58%) quantitative studies [22,23,24,25,26,27,28,29,30,31,32,33,34,35,36,37,38,39,40,41,42,43,44,45,46,47,48,49,50,51,52,53,54,55,56,57,58,59,60,61,62,63,64,65,66,67,68,69,70,71,72,73,74,75,76,77,78,79,80,81,82,83,84,85,86,87,88,89,90,91,92,93,94,95,96,97,98,99,100,101,102,103], 25 (18%) qualitative studies [104,105,106,107,108,109,110,111,112,113,114,115,116,117,118,119,120,121,122,123,124,125,126,127,128], 6 (4%) studies with a mixed-methods design [129,130,131,132,133,134], and 28 (20%) papers categorized as ‘other’ (reviews, book chapters, editorials, letters, protocols) [5,6,8,135,136,137,138,139,140,141,142,143,144,145,146,147,148,149,150,151,152,153,154,155,156,157,158]. Most articles were published between 2015 and February 2021 (n = 97; 69%), 25 articles (18%) between 2010 and 2015, 10 (7%) articles between 2005 and 2010, and only nine (6%) articles between 2000 and 2005. Sixty-three (45%) studies were conducted in Europe (of which 5% were conducted in more than one European country), 51 (36%) in North America, 11 (8%) in Asia, nine (6%) in Australia, and one (1%) in Africa. Five (4%) studies included more than one country across different continents, and one (1%) paper did not report a country of origin.

Studies were carried out across a variety of advanced cancer diagnoses and treatments. The most frequent diagnoses were breast cancer (n = 28; 20%), prostate cancer (n = 23; 16%), lung cancer (n = 16; 11%), melanoma (*n* = 11; 8%) and renal cell carcinoma (n = 10; 7%). In 34 publications (24%), the study sample included several diagnoses. In 25 (18%) studies, patients received exclusively immunotherapy, in 21 (15%) studies chemotherapy, in 16 (11%) studies hormone therapy, and in 16 (11%) studies targeted therapy. Sixty-three (45%) studies reported on patients receiving several types of treatment, or a combination of more than one type of systemic therapy (e.g., chemotherapy and immunotherapy). Quality assessment summary scores ranged from 0.6 to 1 with a median of 0.8, mean score of 0.83, and standard deviation of 0.12. Appendix A includes the individual study characteristics of included qualitative (Appendix A), quantitative (Appendix A) and mixed-methods studies (Appendix A).

### 3.2. Psychosocial Outcomes

The psychosocial outcomes of living with advanced cancer and receiving ongoing systemic treatment from qualitative and quantitative results, respectively, are summarized here. Table 1 provides an overview of the psychosocial themes and highlights the overlap and discordances between reported psychological outcomes from the qualitative and quantitative studies. Quantitative scores reported in individual studies can be found in Appendix A, and were not elaborated on separately for any of the psychosocial themes because of the great variety in terms of study design, study population, and instruments applied for measuring psychosocial aspects.

#### 3.2.1. Uncertainty

Living with uncertainty was a commonly reported consequence of living with advanced cancer receiving long-term systemic treatment across qualitative studies. Uncertainty centered around prognosis, availability of treatment options, and (potential) benefits or side effects of treatment. Patients equated starting (new) treatments with the often-uncertain outcomes of “gambling” and expressed fears about making the wrong treatment decisions. Uncertainty revolved around recurrent medical events such as scans or check-ups that impact the patient’s future perspective. Ongoing uncertainty complicated choices on a medical level as well as on a personal level, such as making plans for the future, leading to feelings of distress, despair, fear, and frustration. Patients experienced a loss of sense of control due to living with uncertainty, and reacted by seeking control in other areas. This included denying the possible negative outcomes of their disease and treatment and holding on to positive thoughts, adopting a healthy lifestyle, living day by day, and making future plans. Additionally, patients sought control in the medical setting by gaining as much medical information as possible in order to make their own treatment decisions, or by seeking trust in their (specialized) hospital, oncologist, and treatment. Another strategy for coping with uncertainty was accepting that the situation involves limited control and uncertainty.

No quantitative studies have reported on uncertainty as a psychosocial outcome.

#### 3.2.2. Fear and Anxiety

In qualitative studies, patients reported fear around three main themes. First, fear of progression of the disease (e.g., because of, or while discontinuing, treatment); second, fear around treatment (e.g., about the potential impact of side effects on daily life, or treatment failure); and third, fear of dying or death (e.g., not feeling prepared for death). Patients applied several strategies to deal with these fears, including avoiding triggers such as talking or thinking about disease and death. They focused on life instead of death, focused on their treatment, and sought distraction by keeping busy and active, for example by exercising or seeking social company. Some patients prepared for their end of life in a practical way by discussing the possibility of end-of-life care or euthanasia with their doctors, or by adapting a spiritual approach for dealing with end of life and death.

Thirty-one of the 82 included quantitative studies (38%) examined anxiety using one or more questionnaires, including the Hospital Anxiety and Depression Scale (HADS) (n = 23; 74%), the Symptom Checklist-90 (SCL-90)–Anxiety subscale (n = 3; 10%), the Hamilton Anxiety rating scale (HAM-A) (n = 2; 6%), the Patient-Reported Outcomes Measurement Information System (PROMIS)–Anxiety questionnaire (n = 1; 3%), the Brief Anxiety Scale (BAS) (n = 1; 3%), the State Anxiety Scale (SAS) (n = 1; 3%), or the General Anxiety Disorder-7 scale (GAD-7) (n = 1; 3%). Three studies reported on fear of progression or recurrence with the Fear of Cancer Recurrence Inventory (FCRI), the Cancer Worry Scale (CWS), the Fear of Progression Short Form (FoP-Q-SF), and a non-validated questionnaire designed by the investigators themselves. Fear of death was examined in one study using a non-validated self-designed questionnaire. Six studies examined (post-traumatic) stress symptoms, and intrusive thoughts and avoidance by using the Impact of Event Scale (IES) or the Post-Traumatic Stress Disorder–Civilian Version questionnaire (PTSD-CV).

#### 3.2.3. Loss

In qualitative studies, patients experienced losses in several domains. Patients experienced losses in their physical health because of their disease and treatment. Limitations in their normal activities, changes in physical appearance, and impaired functioning led to feelings of frustration and distress. Patients were concerned that they would eventually become dependent on others for help to perform daily activities. Additionally, patients experienced losses regarding their perspective on life. Some patients struggled to continue everyday life because they lost the ability to live their life untroubled. Feelings of anger and sadness were a consequence of their non-deliberately changed life, and the loss of their former prospects for the future. Finally, patients were confronted with changes in their ability to work as they did prior to their diagnosis. They had lost or had to give up their jobs, leading to feelings of distress because of loss of income, missing the contact with former colleagues, the lack of a sense of purpose or being of value, and loss of self-esteem. These losses contributed to experiencing the loss of one’s identity.

Patients dealt with their losses in various ways. Several patients mentioned accepting the costs of their disease, treatment, and the prospect of death in the nearer future, thereby accepting their new situation or ‘new normal’. They adapted to the new situation, and readapted when the disease progressed. Patients adjusted by not allowing themselves to dwell on self-pity or negative thoughts. Other coping mechanisms were finding or searching for a (new) meaning in life, for instance by reconsidering life decisions or future plans, finding meaning in supporting fellow patients, or even in their own suffering. Patients also strived to focus on what remained, by trying to maintain normal life as much as possible, enjoying the small things in life and focusing on the here and now. Patients reported trying to be grateful for the time left, and some were even capable of putting their own situation into perspective and experiencing personal growth because of their disease.

No quantitative studies reported on loss as a psychosocial outcome.

#### 3.2.4. Mood and Depression

In the qualitative studies, mood and depression were mentioned in the context of other psychosocial themes, but did not appear to be a prominent psychosocial outcome on its own. For example, patients mentioned feelings of sadness related to the death of fellow patients (related to the theme ‘social life’, see below), feelings of anger because of the loss of their life as it used to be (related to the theme ‘loss’), or frustrations about the lack of (new) treatment options (related to ‘hope and uncertainty’).

A considerable part of the quantitative studies examined symptoms of depression. Twenty-three studies (74%) used the HADS–Depression subscale. Other studies applied one or more of the following questionnaires: the Beck Depression Inventory (BDI), the Montgomery–Asberg Depression Rating Scale (MADRS), the PROMIS–Depression, the Profile of Mood States (POMS)–Depression subscale, the Patient Health Questionnaire-9 (PHQ-9), the SCL-90–Depression subscale, the Center for Epidemiologic Studies–Depression scale (CES-D), or the Self-Rating Depression Scale (SDS). Psychological distress was examined with the Brief Symptom Inventory (BSI), the Distress thermometer and Problem list (DT&PL) and the six-item Kessler Psychological Distress Scale (K6). Mood was examined in one study using the Profile of Moods States (POMS) questionnaire, and one study reported on suicide by analyzing the prevalence of International Classification of Diseases and Related Health Problems (ICD-10) codes.

#### 3.2.5. Hope

Patients also expressed feelings of hope and optimism. Hope was adopted as an adjustment mechanism in order to alter the situation to a more positive one, in order to reduce distress. Patients found hope in positive stories of fellow patients and from peer support. Hope was often expressed in the context of treatment results: maintaining hope for new treatment options, feeling lucky to be able to receive treatment and to benefit from it, hope that treatment would improve quality of life or maintain the possibility of living a normal life, hope that treatment would prolong life, that the cancer would become a chronic condition, or would even be cured. Particularly when receiving trial medication, hope was centered on the unknown positive treatment outcomes and the possibility of further treatment options becoming available in time. Positive treatment results increased feelings of hope, while negative treatment results or disease progression led to increased feelings of hopelessness or complete loss of hope for the future, adding to the continuous struggle between staying hopeful and being realistic or in despair.

Quantitative results involving measures of hope and optimism were scarce: two studies (2%) applied the Mental Adjustment to Cancer scale (MAC), which contains a subscale score assessing helplessness. One study (1%) examined treatment-specific optimism with a self-developed non-validated scale. Another study (1%) applied the Beliefs about Medicine Questionnaire–Specific (BMQ) to report on treatment beliefs in relation to adverse events and permanent medication stoppages.

#### 3.2.6. Social Life

Living longer with advanced cancer and systemic treatment had a profound effect on patients’ social life. Patients found it difficult to deal with the emotions of their loved ones, and experienced feelings of guilt and worry regarding the impact of their illness and future death on loved ones. In addition, feelings of shame occurred, for instance when patients were not able to keep up their work in order to provide for their families. In some cases, patients reported that friends and family found it difficult to understand their condition, which they believed was a consequence of the social or societal misconception that cancer is either curative or terminal. The fact that people do not always look sick on long-term treatments compounded these difficulties. Family, friends, peers, and peer support groups were an important source of support for many patients, although cancer progression or death of fellow patients led to increased feelings of sadness and distress and constituted a confrontation with their own situation. Several patients described losing relationships because of their altered circumstances. Sometimes, patients worried about missing milestones of their loved ones, for instance, attending their grandchildren’s graduation, but at the same time they maintained hope. Patients tried to protect their loved ones from emotional strain, some by minimizing talking about their situation, and others by actively involving and informing loved ones about their disease, treatment, and plans for the phase of dying. For patients with young children, talking about death was considered emotionally challenging. For some, the motivation to continue treatment was to stay alive for their loved ones.

No instruments specifically measuring relationship factors or the social impact were used in quantitative studies.

#### 3.2.7. Quality of Life

In their struggle to weigh length of extended life against quality of life, some patients expressed not feeling ready to die, and having nothing to lose while receiving ongoing systemic treatment. These patients adopted a fighting spirit with the goal of prolonging life. However, not all patients were willing to prolong life at any cost, referring to the life-long uncertainty and the impact of (potential) treatment side effects.

Distinct aspects of quality of life (mental health, emotional well-being, and social belonging) were reported in the qualitative studies, where the majority of the quantitative studies examined general QoL or Health-Related Quality of Life (HRQoL). The European Organization for Research and Treatment for Cancer Quality of Life Questionnaire (EORTC-QLQ-C30) and the Functional Assessment of Cancer Therapy questionnaire (FACT; including the -General (n = 19), FACT-Breast (n = 7), FACT-Prostate (n = 4) and FACT-Lung (n = 2)) were the most frequently applied, in 29 (35%) and 25 (30%) of the 82 quantitative papers, respectively. The EuroQol 5D (EQ-5D; n = 11), the 36- or 12-item Short-Form Health Survey (SF-36; SF-12, n = 6), the Edmonton Symptom Assessment System (ESAS); n = 2), the Assessment of Quality of Life (AQol-8D; n = 1), the Cancer Institute Quality of Life Questionnaire (CI-QOL-Q; n = 1), the Functional Assessment of Chronic Illness Therapy (FACIT; n = 1), or the abbreviated World Health Organisation Quality of Life questionnaire (WHOQoL-BREF; n = 1) were used less frequently. The general QoL baseline mean scores and, if applicable, baseline mean scores for the emotional functioning (EORTC-QLQ-C30) or emotional wellbeing (FACT; SF) subscales were extracted (Appendix A). However, because of the diversity of instruments applied in patients with different diagnoses, the variety of disease stages, and the diverse treatments received, scores were not elaborated upon in this section.

### 3.3. Terminology

From the qualitative studies, we extracted how patients perceived or referred to their disease. Patients mentioned being aware that their treatment was life-prolonging instead of curative, and that death was inevitable in the near future. Some referred to their condition as “living between life and death”, or “between hope and despair”. However, others considered having advanced cancer to be similar to having any other chronic disease, and were not aware of, or not willing to consider, an inevitable death. Disease and treatment were perceived as an inconvenience, and death was not felt as being near. In relation to cancer terminology, patients associated ‘palliative care’ with a shortened lifespan, and some had negative associations with the term ‘terminal’.

The large diversity in terms found in publications used to refer to advanced cancer patients receiving ongoing systemic treatment is summarized in Appendix A. The majority of papers used the terms “advanced”, “metastatic”, or “incurable”, and “cancer” or “disease”, or a combination thereof (e.g., “advanced metastatic cancer”), to indicate the disease stage. Few studies used terms like “late-stage”, “palliative” or “terminally ill” patients. Terms to indicate ongoing systemic treatment or treatment response were “undergoing [type of] treatment/therapy”, “receiving systemic treatment”, “receiving metastatic [type of] cancer treatment”, “long-term responders”, or patients “who achieved a durable response to therapy”. These terms were often combined with terms indicating the disease stage. Several terms specified the permanent status of having to live with the disease, including a life-limiting prognosis, such as “living with life-prolonging therapy”, “living (longer) with metastatic cancer”, “chronic advanced or metastatic cancer”, “life-limiting illness without cure” or “patients with metastatic disease and a limited life expectancy”. Last, several expressions associated with ‘surviving’ were noted, for example, “long-term survivors”, “metastatic cancer survivors”, “metavivorship”, “new survivor population” or “not surviving after but with cancer”.

## 4. Discussion

This scoping review outlined the current literature, including 141 papers on the psychosocial consequences of living long term with advanced cancer and ongoing systemic treatment, and summarized the terminology used to refer to these patients.

Reported psychological outcomes included (1) ongoing uncertainty, (2) anxiety and fear about disease progression, treatment and death, and (3) hope for positive treatment effects and new treatment options, (4) losses in several domains of life, and (5) the disease having a profound impact on their loved ones and social relationships. All these aspects affected general quality of life, and patients experienced feelings of distress and depression with which they tried to cope in several ways.

Uncertainty appeared to be the most prominent concern. Feelings of ongoing uncertainty were perceived with respect to the diagnosis and prognosis, scan results, the potential occurrence of cancer progression, future perspectives, and dying. As observed in this literature overview and confirmed by previous literature, experiencing uncertainty led to feelings of worry, fear, and anxiety [159]. Fear and anxiety, relating to a present or definite danger, or a future or indefinite danger respectively, are common among patients with various chronic diseases, including advanced cancer [160]. Fear of progression occurs in the state of a persistent disease and differs from fear of recurrence which relates to a state where clear episodes of complete remission and subsequent relapse can occur [161]. Our results on advanced cancer patients receiving ongoing treatment indicated fear of progression focused on uncertainty around scans and scan results, potential disease progression, and dying.

The counterpart of feelings of uncertainty and fear in a precarious balance is hope. Hope emerges when a situation is considered to be uncertain [162]. As recently hypothesized by Balen and Merluzzi (2021), uncertainty is perceived to be a major element of hope. In the context of a serious disease such as advanced cancer with high uncertainty regarding the effectiveness of treatments, hope seems to be crucial in handling the situation. As theorized by Herth in 1990 [163], hope is focused on prospects and goals in the future. Our results accordingly showed that patients’ hope was focused on positive treatment effects and the possibility of subsequent treatment options in the case of treatment resistance, as well as achieving future milestones, such as attending the wedding of a loved one, or seeing children or grandchildren growing up. In addition to uncertainty, control is related to hope as well [162]. Control consists of active behavior and internal cognitive processes that an individual can actuate to maintain hope [164]. In the papers included in our study, patients’ active behavior in tackling feelings of lack of control was characterized by seeking control through medical knowledge, adopting a healthy lifestyle, and actively making plans for the future. Seeking control through increasing knowledge was found among various chronic diseases [161], emphasizing the importance of informing patients accurately and completely in order to diminish uncertainty as much as possible. Cognitive processes to gain control were characterized by putting trust in the oncologist and treatment and finding (new) meaning in the challenging situation of advanced illness, sometimes aided by a spiritual approach. In contrast to patients with curative or terminal cancer diagnoses, advanced cancer patients face a changing long-term situation regarding their disease trajectory, leading to continuous uncertainty that calls for hope in order to carry on.

These prominent psychological mechanisms and their interrelationships give directions for healthcare professionals to pay attention to and act upon. Seeking control and providing hope in the management of uncertainty can be guided by doctors and nurses by concretely informing patients about prognosis, treatment possibilities, and waiting times for scan results. Furthermore, counselling of patients during their disease trajectory may be desired for some, for instance, with the aim of assisting patients to accept the aspects of their situation that cannot be controlled, finding a sense of control in aspects of daily life, and helping them to find new meaning in life, using a powerful construct like hope in achieving short-term goals.

Having advanced cancer and receiving ongoing treatment has a profound social impact [165]. Patients mentioned changes in their social life, regularly preceded by the inability of others to fully comprehend the patient’s situation. Adding to this is the judgement of others with respect to patients’ appearance or functioning, which has been reported in the previous literature as well [165], and patients described the stigmatization of others, assuming that they must do and/or feel good when they appear to look good or were able to keep up normal activities such as going to the sports club or on holidays. In general, patients found support in social contacts, which helped them to hold on to future milestones with loved ones, or continuing treatment. For some patients, peer support was of great added value. However, some patients tend to compare themselves with fellow patients that have more severe disease or have experienced negative treatment outcomes. Previous research shows that this downward comparison increases uncertainty and fear of progression, leading to worse health states, including increased feelings of distress [161].

Moreover, comparable to what has been found in patients with other chronic diseases, a significant concern for patients was the impact of the disease on their loved ones, including the emotional burden caused, and having to leave their loved ones behind [161]. This was most apparent for close family members, such as partners and children. It is, therefore, important to actively involve partners in the disease trajectory and to regularly check in on their mental functioning and needs as well. Future research on the patient–caregiver relationship could also provide insight into this dyadic process and can assist in the development of tools for support. Within this research, it is important to include the caregiver perspective, as their concerns might be different from what is reported by patients. Furthermore, to prevent social isolation, patients and partners could benefit from support on how to communicate about their partner’s disease and situation with others, in order to gain a feeling for what they can control in this matter and how to cope with misunderstandings.

Given the long-term, continually changing situation regarding the patient’s disease trajectory, response shift is likely to occur. Response shift is a psychological process defined by Sprangers and Schwartz (1999) as “the change in an individual’s internal standards, values or conceptualisations that occurs in response to a particular catalyst such as illness” [166]. It delineates people’s adjustment in terms of expectations and/or goals in order to accommodate their changed circumstances, which is necessary for maintaining a good quality of life. Advanced cancer patients are recurrently challenged to adjust to their changing circumstances, for example, when treatment fails or disease progression occurs, which continuously threatens or alters their quality of life. This aligns with the losses that patients described in almost every domain of life. Response-shift was observed as repetitively readapting by finding or searching for a (new) meaning in life, and reconsidering life decisions or future plans. Patients also strived to focus on what remained, enjoying the small things in life and focusing on the here and now.

How well patients are able to cope might vary among people and through time. Although the majority of quantitative studies measured QoL as psychosocial outcome, it is not clear how QoL changes over time in an ongoing changing situation of a life-long progressive disease. However, in the case of a life-limiting disease, maintaining quality of life is a central goal, and the psychosocial, emotional, and spiritual components of QoL seem to be significant contributors to wellbeing. Although physically incapacitated, patients can still experience a high quality of life [167]. While these QoL components have been addressed separately in qualitative research, several quantitative studies, including most clinical trials focused on measuring general quality of life, emphasized the physical scales or global QoL scores [168]. Furthermore, psychosocial aspects have often been examined by applying general instruments instead of disease-specific instruments. A benefit of this approach is that it improves the comparability between different studies and different patient groups in the previous literature, but it adds little to the increasing knowledge of specific psychosocial outcomes relevant for advanced cancer patients receiving long-term systemic treatment. Together with the issue of response shift, which is likely to occur more frequently in advanced cancer patients, this calls for research focused on a longer-term examination of quality of life and psychosocial mechanisms. Specific instruments on uncertainty, fear, hope and loss, validated for this patient population, are required to gain insight in the prevalence, predictors, and course of psychosocial consequences of living long-term with advanced cancer.

In this scoping review, patients with varying advanced cancer diagnoses and diverse systemic treatment types, whether or not combined with non-systemic therapies like radiotherapy or surgery, were examined. The great variety in terminology used to refer to patients with advanced cancer receiving ongoing systemic treatment complicates research, as it is difficult to survey the literature in this field and to compare research results. To increase knowledge of the specific (psychosocial) needs of this patient group, uniformity and consensus on terminology is vital. Although some patients advocate terms as survivors(hip) or metavivors [6,8], others do not recognize themselves in these terms, as they might implicate absence of disease. These terms might be more applicable to those who achieve complete remission after immunotherapy (e.g., some melanoma patients). For patients with active disease who are benefitting from systemic treatments, a term like ‘long-term responders’ could be more suitable and helpful for research purposes once codesigned or verified with relevant stakeholders such as patients, caregivers and others involved. We therefore call for a broad discussion and consensus on terminology to be used in future studies, to be able to move this growing field of research forward effectively.

In the absence of a universal term, our extensive search strategy (Appendix A, Appendix A) combined terms on advanced disease, systemic treatments, and psychosocial outcomes. This resulted in the inclusion of a relatively large number of quantitative studies examining systemic treatments and psychosocial outcomes such as global QoL, whereas (qualitative) studies that did not mention treatment-related terms in the title, abstract or keywords might have been missed. On the other hand, leaving treatment terms out of our search was not feasible, as it led to a multiplication of the number of articles found primarily. A strength of this study is the wide scope of information on psychosocial outcomes across heterogeneous study populations (cancer diagnoses and treatments), study types, and applied instruments. Although the majority of quantitative studies used the EORTC-QLQ-C30 or FACT questionnaires to measure QoL, a meta-analysis of the quantitative results was beyond the scope of our study, and an overall conclusion on quantitative results was therefore not given. While our study provides a useful overview of common psychological themes in advanced cancer, future research is needed into the specific challenges and needs for subgroups of patients related to different treatment modalities. For instance, targeted therapies might be continued when effective, while immunotherapy might be discontinued after a certain period based on individual response. Insight into these unique challenges will assist in developing tailored psychosocial support.

## 5. Conclusions

Due to the rapid developments in systemic treatment options improving progression-free and overall survival rates, an increasing number of patients with various advanced cancer diagnoses will receive more lines of life-prolonging systemic treatment for longer periods of time. Among 141 studies, the qualitative studies showed that these patients deal with uncertainty, fear, hope, loss, and worry about social life. As these specific issues were not studied extensively in quantitative studies, more research is needed on the prevalence, severity, and course of the psychosocial outcomes, as well as on theoretical models for future interventions in this specific patient group. We recommend the use of validated instruments to quantify specific psychosocial outcomes relevant for this newly recognized distinct group of locally advanced and metastatic cancer patients receiving novel and long-term systemic treatments, guided by suitable terminology in accordance with patients, researchers, and health care professionals.

## Figures and Tables

**Table 1 cancers-14-03889-t001:** Overview of psychosocial themes, concepts, and instruments.

Themes in Qualitative Studies	Concepts in Quantitative Studies	Quantitative Instruments
**1. Uncertainty**		
Uncertainty	Not reported	Not quantified
**2. Fear and anxiety**		
Fear of cancer progression	Fear of disease progression	FoP-Q-SF [63]
Not reported	Fear of cancer recurrence	FCRI(-SF) [36,63]; CWS [36]
Fear of dying	Fear of death	Non-validated instrument [59]
Anxiety	Anxiety	BAS [27]; GAD-7 [62]; HADS–Anxiety [27,42,46,47,48,53,58,64,66,67,68,79,80,81,82,83,86,87,89,90,91,94,99]; HAM-A [46,48]; PROMIS–Anxiety [29]; SAS [102];
Not reported	Intrusive thoughts	IES [23,24,36,41,48]
Not reported	(Post-traumatic) stress	PTSD-CV [88]
**3. Loss**		
Loss	Not reported	Not quantified
**4. Mood and depression**		
Not identified as separate theme but reported in relation to other psychosocial themes	Mood	POMS-TMD subscale [89]
Not identified as separate theme but reported in relation to other psychosocial themes	Depression	MADRS [27,48,69]; BDI [22,27,69,84,100]; HADS–Depression; [27,42,46,47,48,53,58,64,66,67,68,79,80,81,82,83,86,87,89,90,91,94,99]; PROMIS–Depression [29]; POMS–Depression [35]; PHQ-9 [46,56,62]; CES-D [28,35,102]; SDS [49]
Not identified as separate theme but reported in relation to other psychosocial themes	(Psychological) distress	BSI [35,41]; DT&PL [66,86]; K6 [132]; SCL-90 [22,27,97]
Not reported	Suicide	ICD-10 [78]
**5. Hope**		
Hope	Helplessness and hopelessness	MAC subscale hopelessness [57,89]
Not identified as separate theme but reported in relation to other psychosocial themes	Optimism	LoT-R [76]
Not reported	Treatment-specific optimism	Non-validated instrument [35]
**6. Social life**		
Impact on social life	Not reported	Not quantified
**7. (Health-related) Quality of Life ((HR)QoL)**		
Several aspects of (HR)QoL were reported in relation to other psychosocial themes	(HR)QoL	AQoL-8D [77]; CI-QOL-Q [96]; EORTC-QLQ-C30 [25,26,30,31,32,36,38,43,51,55,57,60,61,64,72,75,79,85,86,87,91,92,94,95,99,100,130,133,134]; EORTC-QLQ-C15-PAL [64]; EQ-5D [33,34,38,44,51,55,61,62,93,98,103,129]; ESAS [58,71]; FACIT [73]; FACT [28,29,37,39,40,42,44,45,46,47,48,50,54,56,58,64,71,82,83,90,93,98,101,129,132]; SF-12 [73]; SF-36 [23,24,70,74,88,102]; WHOQoL-BRIEF [103]
**8. Cognitive aspects/beliefs**		
Not reported	Treatment-related regret	DRS [52]
Not reported	Decisional conflict	DCS [52]
Not identified as separate theme but reported in relation to other psychosocial themes	Treatment beliefs	BMQ [149]
**9. Coping/adjustment**		
Not identified as separate theme but reported in relation to other psychosocial themes	Coping strategies	MCMQ [57]
Not reported	Mental adjustment to cancer	MAC [57,89]
Not identified as separate theme but reported in relation to other psychosocial themes	Acceptance	ICQ-acceptance [80,81]
Not reported	Self-efficacy	SES [79]; GSE [132]
Not identified as separate theme but reported in relation to other psychosocial themes	Spirituality	FACIT-SP [37]

FoP-Q-SF: Fear of Progression Questionnaire–Short Form; FCRI-SF: Fear of Cancer Recurrence Inventory–Short Form; CWS: Cancer Worry Scale; BAS: Brief Anxiety Scale; GAD: General Anxiety Disorder; HADS: Hospital Anxiety and Depression Scale; HAM-A: Hamilton Anxiety Rating Scale; PROMIS: Patient-Reported Outcomes Measurement Information System; SAS: State Anxiety Scale; IES: Impact of Event Scale; PTSD-CV: Post-traumatic Stress Disorder- Civilian Version; POMS: Profile of Mood States; MADRS: Montgomery–Asberg Depression Rating Scale; BDI: Beck Depression Inventory; SCL-90: Symptom Check List-90; EQ-5D: EuroQol-5D; PHQ: Patient Health Questionnaire; CES-D: Center for Epidemiology Studies Depression Scale; FACT: Functional Assessment of Cancer Therapy; SF: Short Form Health Survey; MAC: Mental Adjustment to Cancer; SDS: Self-Rating Depression Scale; BSI: Brief Symptom Inventory; DT&PL: Distress Thermometer and Problem List; GSE: General Self-Efficacy Scale; K6: Kessler 6; ICD-10: International Classification of Diseases, 10th Revision; LOT-R: Life Orientation Test–Revised; EORTC-QLQ: European Organisation for Research and Treatment; AQoL: Assessment of Quality of Life; CI-QoL: Cancer Institute–Quality of Life Questionnaire; ESAS: Edmonton Symptom Assessment Scale; FACIT: Functional Assessment of Chronic Illness Therapy; WHOQoL-BRIEF: World Health Organization Quality of Life Brief Questionnaire; DRS: Decisional Regret Scale; DCS: Decisional Conflict Scale; BMQ: Beliefs about Medicine Questionnaire (BMQ); MCMQ: Medical Coping Modes Questionnaire; ICQ: Illness Cognition Questionnaire; SES: Self-Efficacy Scale.

## Data Availability

Further data is available upon request.

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
