# Peer review of "Psychosocial Aspects of Living Long Term with Advanced Cancer and Ongoing Systemic Treatment: A Scoping Review"

_cancers, 2022, doi:10.3390/cancers14163889_

Round 1

Reviewer 1 Report

Dear Authors,

The paper presents a topic of interest for researchers and practitioners. However, several improvements are needed.

1. The abstract should present the need for research, the methodology, the main results obtained and the future directions of research.

2. The introduction should be supplemented with other research conducted in this field and with other developed models developed. In this way the research will show what is the gap it fills and what are the elements of originality.

3. To emphasize the need for this study.

4. The stages of the methodology must be presented in detail.

5. To highlight the gaps filled by the present study.

6. The conclusions section should be completed with a review of the study.

Best regards,

Reviewer

Reviewer 2 Report

The review was well structured using the correct methodology. The research question and specific goals related to the broad overview were well addressed. As the review is rather long, we would recommend omitting the first paragraph of the introduction and figure 1 (results section) is also rather redundant. In the discussion section, it may be useful to look more closely at the patient-caregiver relationship and to suggest specific instruments that this research group would recommend for future research into long-term QOL and other psychosocial mechanisms.

To conclude, I would like to congratulate the authors on this fine summary of psychosocial aspects for "long-term responders".

Round 2

Reviewer 1 Report

I accept this version of the paper.